# Cytostatic Effect of a Novel Mitochondria-Targeted Pyrroline Nitroxide in Human Breast Cancer Lines

**DOI:** 10.3390/ijms22169016

**Published:** 2021-08-20

**Authors:** Kitti Andreidesz, Aliz Szabo, Dominika Kovacs, Balazs Koszegi, Viola Bagone Vantus, Eszter Vamos, Mostafa Isbera, Tamas Kalai, Zita Bognar, Krisztina Kovacs, Ferenc Gallyas

**Affiliations:** 1Department of Biochemistry and Medical Chemistry, University of Pecs Medical School, 7624 Pecs, Hungary; andreidesz.kitti@pte.hu (K.A.); aliz.szabo@aok.pte.hu (A.S.); dominika.kovacs@aok.pte.hu (D.K.); balazs.koszegi@aok.pte.hu (B.K.); viola.vantus@aok.pte.hu (V.B.V.); eszter.vamos@aok.pte.hu (E.V.); zita.bognar@aok.pte.hu (Z.B.); krisztina.kovacs@aok.pte.hu (K.K.); 2Institute of Organic and Medicinal Chemistry, Faculty of Pharmacy, University of Pecs, 7624 Pecs, Hungary; mostafaisbera@gmail.com (M.I.); tamas.kalai@aok.pte.hu (T.K.); 3Szentagothai Research Centre, University of Pecs, 7624 Pecs, Hungary; 4HAS-UP Nuclear-Mitochondrial Interactions Research Group, 1245 Budapest, Hungary

**Keywords:** MDA-MB-231, MCF7, mitochondrial membrane potential, mitochondrial energy metabolism, reactive oxygen species, invasive growth, Mito-CP

## Abstract

Mitochondria have emerged as a prospective target to overcome drug resistance that limits triple-negative breast cancer therapy. A novel mitochondria-targeted compound, HO-5114, demonstrated higher cytotoxicity against human breast cancer lines than its component-derivative, Mito-CP. In this study, we examined HO-5114′s anti-neoplastic properties and its effects on mitochondrial functions in MCF7 and MDA-MB-231 human breast cancer cell lines. At a 10 µM concentration and within 24 h, the drug markedly reduced viability and elevated apoptosis in both cell lines. After seven days of exposure, even at a 75 nM concentration, HO-5114 significantly reduced invasive growth and colony formation. A 4 h treatment with 2.5 µM HO-5114 caused a massive loss of mitochondrial membrane potential, a decrease in basal and maximal respiration, and mitochondrial and glycolytic ATP production. However, reactive oxygen species production was only moderately elevated by HO-5114, indicating that oxidative stress did not significantly contribute to the drug’s anti-neoplastic effect. These data indicate that HO-5114 may have potential for use in the therapy of triple-negative breast cancer; however, the in vivo toxicity and anti-neoplastic effectiveness of the drug must be determined to confirm its potential.

## 1. Introduction

Mitochondria have become novel targets for anti-cancer strategies [1]. While the Warburg effect states that due to defective oxidative phosphorylation, the rate of glycolysis is elevated to replace ATP loss [2], oxidative phosphorylation has been recently recognized to play an important role in oncogenesis. Furthermore, the mitochondria of cancer cells can alternate between glycolysis and oxidative phosphorylation to meet the metabolic demands of the cell and to promote survival [3]. Targeting the mitochondria shows great promise to enhance the efficiency of anti-cancer drugs. Additionally, targeting the mitochondria could mitigate treatment resistance, another crucial factor of today’s anti-cancer therapy. Mitochondria-targeted nanocarriers and drugs conjugated to mitochondria-targeting ligands are the most common approaches [4].

Lipophilic cations, such as tryphenylphosphonium (TPP), are frequently used conjugates in the design of mitochondria-targeted anti-cancer drugs, and they also have antifungal, antiparasitic, and antioxidant uses. The chemical background of mitochondrial targeting by lipophilic cations, such as TPP, is that the delocalized positive charge enables the drug to easily permeate lipid bilayers, which is an advantage compared to hydrophobic compounds that should rely on tissue-specific carriers. Lipophilic cations can achieve efficient uptake and accumulation several hundredfold within the mitochondria depending on the mitochondrial membrane potential (−150 to −180 mV) [5,6]. The mitochondrial membrane potential (ΔΨ_m_) of cancer cells is higher compared to their cytosol and to non-cancer cells, and hence a selective targeting can be achieved [7]. After administration, more than 90% of the intracellular lipophilic cations was found to be located in the mitochondria [6]. When administered orally, the uptake of TPP-based drugs by the mitochondria was fast and organ-selective; they accumulated within the heart, skeletal muscle, liver, and brain of mice [6,8].

Recently, the bioenergetics of cancer cells is receiving increased interest among researchers in the field [9,10]. Breast cancer cells have profound bioenergetic, histological, and genetic differences compared to normal cells [10]. As triple-negative breast cancer (TNBC) represents about 15–20% of all cases and is associated with a poor prognosis and limited therapeutic options, the development of novel therapeutic means is needed [11]. Targeting metabolism and the mitochondria could be a useful therapeutic approach in TNBC cases because mitochondria play pivotal roles in early relapse and the metastatic spread of TNBC. Previous research has also demonstrated that targeting glycolysis might not be an effective strategy in TNBC therapy and has suggested that the mitochondrial aid-in-reserve must be selectively blocked [10].

Mito-CP, a TPP conjugated superoxide dismutase mimetic, was the first mitochondria-targeted nitroxide compound. It was used for studying the role of the mitochondrial superoxide in cancer cell proliferation [12]. Mito-CP has shown cytotoxic properties in various cancer cells, including breast cancer cells, without markedly affecting non-cancerous ones [13,14]. Recently, a novel component-derivative of Mito-CP, a pyrroline nitroxide attached dyphenylphosphine compound, HO-5114, was synthesized [15], which demonstrated markedly higher cytotoxicity against TNBC and hormone receptor positive human breast cancer (HR+BC) lines than Mito-CP. This report presents a detailed study of HO-5114′s effect on the breast cancer lines and provides evidence that the mitochondrial effects of the drug could participate in its cytotoxic and anti-proliferative effects.

## 2. Results

### 2.1. Effect of HO-5114 on Cell Viability

To assess its anti-neoplastic potential, we treated TNBC MDA-MB-231 and HR+BC MCF7 lines with 1, 2.5, 5 or 10 µM HO-5114 for 24 h and then determined their viability using the sulforhodamine B (SRB) assay. The SRB assay measures protein content that is considered to be more proportional to the cell count than metabolic activity, which can under- or over-estimate the cell count if the studied substance inhibits or uncouples mitochondrial oxidative phosphorylation [16]. We found that HO-5114 decreased viability in both breast cancer lines in a concentration- and time-dependent manner (Figure 1). Even at the lowest concentration tested, the drug substantially reduced the viability of both cell lines. In agreement with the view that TNBC is more chemotherapy-resistant than HR+BC, the MDA-MB-231 cells were more resistant against HO-5114 treatment than the MCF7 cells, although the treatment with 10 µM HO-5114 reduced viability below 10% in both cell lines (Figure 1).

### 2.2. Determination of the Type of HO-5114-Induced Cell Death

We determined the type of HO-5114-induced cell death using flow cytometry. The cells were treated exactly as for the viability measurement, and then they were double-stained with fluorescein isothiocyanate (FITC) conjugated Annexin V and propidium iodide (PI). The latter enters the cell if the plasma membrane is disrupted, binds to the double-stranded DNA, and becomes intensely fluorescent, indicating necrosis. The former binds to phosphatidylserine, a marker of apoptosis when it is on the plasma membrane’s outer layer. Double positivity indicates late apoptosis. In MCF7 cells, HO-5114 treatment increased the ratio of early—and to a much higher extent—late apoptotic cells on the expense of live cells in a concentration-dependent manner in the whole concentration range tested (Figure 2a,c). In contrast, <5 µM concentrations of HO-5114 did not have a significant effect on the MDA-MB-231 cells; however, 10 µM of HO-5114 had a pronounced effect. It lowered the live cell ratio to 25% while increasing the ratio of early and late apoptotic cells to 10% and 65%, respectively (Figure 2b,d).

### 2.3. Effect of HO-5114 on Reactive Oxygen Species (ROS) Generation

In many cases, anti-neoplastic agents induce ROS production in cancer cells [17]. Accordingly, we studied HO-5114-induced ROS production in human breast cancer lines using the dihydrorhodamine 123 assay. The assay is based on measuring the fluorescence of rhodamine 123 produced quantitatively from its non-fluorescent reduced form by the cellular ROS. At a 10 µM concentration, which lowered the viability of both human cancer lines to less than 10% of that of the untreated control, HO-5114 caused ROS production to the extent of about 1.7 and 2 times of the untreated control in the TNBC and HR+BC lines, respectively (Figure 3). Treatment with lower concentrations of the drug that still induced a massive decrease in the viability of both cell lines caused no or only slight cellular ROS production (Figure 3), suggesting that the induction of oxidative stress was unlikely involved among the mechanisms of HO-5114′s cytotoxicity. Increasing HO-5114′s concentration to 20 µM elevated ROS production proportionally in both cell lines (Figure 3), indicating that ROS production was likely far from the saturation level under these conditions.

To investigate the role of oxidative stress induction in the anti-neoplastic effect of HO-5114, we studied how an antioxidant affects HO-5114′s cytotoxicity in BC cells. To this end, we treated the MCF7 and MDA-MB-231 cells with 1, 2.5, 5 or 10 µM HO-5114 for 24 h in the presence or absence of 1 mM N-acetylcysteine (NAC) and then measured the viability using the SRB assay. We could not observe any effect of NAC on HO-5114′s cytotoxicity in the case of the HR+BC line MCF-7 (Figure 4a). In contrast, in the TNBC line MDA-MB-231, NAC significantly increased the viability of the control cells as well as the cells treated with up to 5 µM HO-5114; however, at a 10 µM HO-5114 concentration, there was no difference in viability between cells treated in the presence and absence of NAC (Figure 4b).

### 2.4. Effect of HO-5114 on ΔΨ_m_

HO-5114 is targeted to the mitochondria due to its diphenylphosphonium component. Therefore, we studied whether it affects ΔΨ_m_ by measuring the JC-1 fluorescence. Based on its cationic properties, JC-1 is taken up by the mitochondria in a ΔΨ_m_-dependent manner. In healthy mitochondria, it forms red fluorescent J-aggregates. Mitochondrial damage results in decreased ΔΨ_m_, leading to a lower accumulation of JC-1 in the form of green fluorescent monomers, while the fluorescence disappears when the ΔΨ_m_ dissipates completely. After merely a 1 h treatment, HO-5114 at the concentration of 1 µM caused a significant drop in the ΔΨ_m_ of MCF7 cells, while increasing the drug’s concentration to 2.5 µM resulted in a massive ΔΨ_m_ loss indicated by the almost complete disappearance of the red fluorescence of JC-1 (Figure 5a,c). The MDA-MB-231 line was more resistant to HO-5114; the same concentrations triggered basically the same changes in the ΔΨ_m_ that were observed for the MCF7 cells, but it necessitated 2.5 h of treatment rather than 1 h only (Figure 5b,d).

### 2.5. Effect of HO-5114 on Mitochondrial Energy Production

Due to the increasing importance of energy metabolism among the pathomechanisms of cancer [3], we studied the effect of HO-5114 on the mitochondrial energy production of MDA-MB-231 and MCF7 lines using the Seahorse XFp Cell Mito Stress Test Kit. The device simultaneously measures the real-time cellular oxygen consumption rate (OCR) and extracellular acidification rate (ECAR), indicators of mitochondrial respiration and aerobic glycolysis, respectively. The cells were treated with 1 or 2.5 µM HO-5114 for 4 h, while OCR and ECAR were monitored during the last 75 min of treatment. Basal respiration was recorded for 15 min (Figure 6a; 1), and then the F_o_F_1_ ATPase inhibitor oligomycin was administered to assess ATP production (Figure 6a; 4). After another 20 min of recording, mitochondrial electron transport and ATP synthesis were uncoupled from each other by adding carbonyl cyanide 4-(trifluoromethoxy) phenylhydrazone (FCCP) to determine maximal respiration (Figure 6a; 3). After an additional further 20 min of recording, mitochondrial respiration was blocked by adding rotenone and antimycin A, inhibitors of Complex I and III of the mitochondrial respiratory chain, to determine proton leak and non-mitochondrial oxygen consumption (Figure 6a; 2 and 5).

From the recorded raw data (Figure 6a,b), the Seahorse instrument generated multiple parameters of cellular energy metabolism (Figure 6c–i) that were all diminished by HO-5114 treatment except the proton leak, which was not affected in either cell line (Figure 6f). Furthermore, coupling efficiency that indicates how tightly respiration is coupled to ATP synthesis was not affected in the MCF7 line but was decreased in the MDA-MB-231 line (Figure 6i). The parameters of cellular energy metabolism associated with mitochondrial oxygen consumption, such as basal respiration, maximal respiration, and ATP production, were lower in the TNBC cells than in the HR+BC cells. Furthermore, 1 and 2.5 μM HO-5114 decreased these parameters to about the same extent for the latter cell line, while it affected them in a concentration-dependent manner for the former (Figure 6d,e,g). Administration of the ATP synthesis inhibitor oligomycin diminished OCR, which was accompanied by an elevation in ECAR in both cell lines (Figure 6j,k). HO-5114 at a concentration of 1 and 2.5 μM reduced ECAR to about the same extent in the MCF7 line, while it increased and decreased ECAR compared to the untreated control at 1 and 2.5 μM, respectively (Figure 6j,k).

Similar to the viability studies, we investigated the effect of NAC on the energy metabolism of untreated and HO-5114-treated BC cells. To this end, we included 1 mM NAC in a set of HO-5114-treated cells throughout the experiment. In the presence of NAC, the effect of HO-5114 on all parameters of cellular energy metabolism except the proton leak was reversed, in a higher extent for the MDA-MB-231 line than for the MCF7 line (Figure 7). In the MCF7 cells, HO-5114 decreased the proton leak that was further decreased in the presence of NAC. In contrast, HO-5114 increased the proton leak of the MDA-MB-231 cells that was further increased in the presence of NAC (Figure 7f).

### 2.6. Effect of HO-5114 on Colony Formation

A colony formation assay was performed to assess the proliferation capacity of MCF7 and MDA-MB-231 cells treated with different concentrations of HO-5114. The cells were cultured in the presence of 50, 75, 100 or 250 nM of HO-5114 for seven days, and then the colonies were stained and counted. The drug effectively reduced colony formation in a concentration-dependent manner in both cell lines (Figure 8). Interestingly, the TNBC line was more sensitive to the treatment than the HR+BC line; 250 nM HO-5114 completely eradicated the MDA-MB-231 cells, while it allowed the survival of about 10 colonies of MCF7 cells (Figure 8).

### 2.7. Effect of HO-5114 on Invasive Growth

Cell proliferation, migration, and invasion are important in understanding tumor progression and metastasis formation [18]. We used the xCELLigence Real-Time Cell Analysis method to assess the effect of HO-5114 on the invasive growth characteristics of MCF7 and MDA-MB-231 cells. The instrument measures electron flow transmitted between gold microelectrodes fused to the bottom surface of a microtiter plate in the presence of an electrically conductive culturing medium. Adherent cells cultured in the plates change the impedance expressed as arbitrary units called the cell index, the magnitude of which is dependent on number, morphology, size, and attachment properties of the cells. The cells were cultured in the presence of 75, 100 or 250 nM of HO-5114 for seven days, while the cell index was monitored in real-time. The drug effectively reduced the cell index in a concentration-dependent manner in both cell lines (Figure 9). At the highest concentration (250 nM), HO-5114 decreased invasive growth close to the detection limit in both cell lines. Similar to the colony formation experiments, the TNBC line was more sensitive to the treatment than the HR+BC line (Figure 9).

## 3. Discussion

TNBC is considered to have a poorer prognosis and a more limited targeted therapy repertoire than the HR+ subtype [19]. Additionally, the energy metabolism of the two breast cancer subtypes differs profoundly, which is indicated by the opposite effect of mitochondrial rescue on glycolytically inhibited HR+BC and TNBC cells; it is negative for the former and positive for the latter [10]. Accordingly, mitochondria-targeted compounds that compromise mitochondrial energy production may prove effective in the therapy of TNBC [13]. Mito-CP was reported to deplete the cellular ATP level, to inhibit mitochondrial oxygen consumption, to affect mitochondrial morphology, and to dissipate ΔΨ_m_ [14]. As a component-derivative of Mito-CP [15], HO-5114 was expected to have similar mitochondrial effects. The drug exceeded these expectations because 10 µM of HO-5114 suppressed viability to about the same extent as 50 µM Mito-CP during a 24 h exposure [15]. In complete agreement with these previous results, in the present study, we found that even 1 µM of HO-5114 decreased the viability of both human breast cancer lines by more than 35%, while it almost completely suppressed it at a 10 µM concentration (Figure 1). At a longer exposure time (48 h), the drug’s anti-proliferative effect became more pronounced in both the HR+ and the TNBC lines (Figure 1).

Mitochondria affect cancer cell survival through at least three major mechanisms: energy production, the intrinsic apoptotic pathway, and ROS generation [20]. These three pathways are interrelated because apoptosis is an energy-dependent process, while energy shortage and the resulting decrease in ΔΨ_m_ leads to the release of pro-apoptotic intermembrane proteins, such as cytochrome c, an apoptosis-inducing factor, and endonuclease G [21]. ROS damages the mitochondrial electron-transport chain and thus the ATP production, while the compromised electron-transport chain produces more ROS [2]. ROS activates apoptosis via damaging macromolecules and interfering with the pro-apoptotic signaling pathways [22]. We observed a substantial induction of apoptosis after a 24 h exposure to 10 µM of HO-5114 in the TNBC line, while lower concentrations of the drug were ineffective in this respect (Figure 2b). In contrast, and in full agreement with the widely accepted view that TNBC is more apoptosis-resistant than HR+BC [11], even 1 µM of HO-5114 induced massive apoptosis in the MCF7 line (Figure 2a).

ROS participates in mediating cancer phenotype remodeling that manifests in apoptosis resistance and increased metastatic properties [23]. The chronic hypoxia prevalent in solid tumors results in the constant activation of the hypoxia-inducible factor-1α transcription factor that induces a malignant transformation associated metabolic remodeling [24]; however, we found a very similar extent of HO-5114-induced ROS formation in the MCF7 and MDA-MB-231 lines (Figure 3), although the latter represents a higher stage of metabolic transformation than the former [23]. The moderate increase in ROS accumulation in response to an increased HO-5114 concentration to 20 µM (Figure 3) also indicated that ROS production in the BC lines was insensitive to HO-5114 treatment, contrary to the expectation. The difference in conditions between solid tumors and the cell culture, where uniform oxygen and fuel supply is provided, may account for the discrepancy between the expected and observed ROS production. Elevated ROS production is considered to be necessary for survival and growth of TNBC cells [25], therefore, antioxidants are expected to hinder their survival [26]. However, we found that the antioxidant NAC increased the viability of control MDA-MB-231, while it did not affect MCF7 cells, indicating a higher ROS level that impeded proliferation in the former (Figure 4). The viability promoting effect of NAC overcompensated for the cytotoxic effect of HO-5114 at the concentration of up to 5 µM, but at 10 µM, it failed to do so (Figure 4b). The absence of NAC’s effect on HO-5114′s cytotoxicity in the HR+BC line (Figure 4a) indicated not only a reduced chronic oxidative stress in it compared to the TNBC line but also suggested differences in metabolic reprogramming between the two BC cell lines [27].

The driving force for ATP synthesis is provided by ΔΨ_m_; however, it has additional essential roles, such as transporting nuclearly encoded mitochondrial proteins [28], transporting K^+^, Ca^2+^, and Mg^2+^ [29], generating ROS [30], mitochondrial quality control [31], and the regulation of pro-apoptotic intermembrane protein release [32,33,34]. Cell survival essentially relies on the maintenance of ΔΨ_m_. Accordingly, in ischemic situations, the F_o_F_1_ ATPase can operate in reverse mode and consume ATP to maintain ΔΨ_m_ to rescue the cell. The ATP is supplied by the substrate-level phosphorylation of non-glucose substrates under these conditions; however, considering the amount of the available non-glucose substrate pool, this survival attempt is often futile [35,36,37]. In solid tumors, the cancer cells must adapt their metabolism to the chronic hypoxia and partially ischemic situation [38,39]. In contrast to ROS induction, we observed a very sensitive response of ΔΨ_m_ loss to HO-5114 treatment. Even 1 µM of the drug induced significant changes in ΔΨ_m_ during as short a treatment as 1 h for the MCF7 line and 2.5 h for the MDA-MB-231 line (Figure 5).

Cancer cells face a double challenge in producing enough energy and a sufficient metabolic intermediate for proliferation in a predominantly hypoxic and partially ischemic environment [40]. Mostly, they rely on glycolysis rather than mitochondrial oxidative phosphorylation, even if sufficient oxygen is available for the latter [41]. Accordingly, increased glucose uptake is a characteristic feature of tumors that is used to identify them by ^18^F-deoxyglucose positron emission tomography [42,43] for diagnostic purposes. On the other hand, the most malignant cancer types, such as metastatic tumor cells, therapy-resistant tumor cells, and cancer stem cells, rely on mitochondrial ATP synthesis [44,45]. The survival, proliferation, and metastasis of these cells depend on the oxidative phosphorylation and form the basis of their therapy resistance [46,47]. Accordingly, for the most malignant cancer types, oxidative phosphorylation is an emerging therapeutic target [48], and drugs significantly affecting tumor cell metabolism may have therapeutic value [38]. Considering its effects on energy metabolism in human breast cancer lines, HO-5114 fulfills this criterion. At a 1 and 2.5 µM concentration, it significantly diminished all OCR-related parameters in both cell lines except coupling efficiency (Figure 6). HO-5114 at a 2.5 µM concentration reduced ATP production that could contribute to the drug’s anti-metastatic property. In complete agreement with its effect on the viability of BC lines, NAC counteracted the inhibitory effect of HO-5114 on the various parameters of cellular energy metabolism except the proton leak (Figure 7). These data support the conclusion that HO-5114 affects the energy metabolism of the BC lines. The proton leak can indicate damage to the mitochondrial respiratory chain or regulation of mitochondrial ATP synthesis via uncoupling proteins (UCPs) [49]. Indeed, the role of UCP2 in regulating the balance between substrate-level and oxidative phosphorylation has recently been reported [50]. We found that both BC lines increased ECAR, i.e., substrate-level phosphorylation when oxidative ATP production was blocked by oligomycin (Figure 6 and Figure 7). ECAR in the MDA-MB-231 line even returned to its initial rate when the oxidative phosphorylation was uncoupled by FCCP (Figure 6 and Figure 7), demonstrating that the balance between the two ATP producing machinery is more responsive in the TN than in the HR+BC cells.

The hormone receptor status determines the cell proliferation, differentiation, and cancer progression properties of breast cancers [51]. Accordingly, the MDA-MB-231 line represents a more aggressive, apoptosis- and therapy-resistant phenotype than the HR+ MCF7 line. The results of the aforementioned experiments that involved 1–24 h exposure to HO-5114 were in line with this view; however, in the colony formation (Figure 8) and invasive growth (Figure 9) experiments, where the cells were exposed to a 50–250 nM concentration of the drug for seven days, MDA-MB-231 proved to be more sensitive to the treatment than the MCF7 line. The reason for this difference in sensitivity to HO-5114 treatment between short- and long-term exposure is not clear based on the experiments.

In conclusion, all data acquired in this study indicated that HO-5114 had a robust anti-neoplastic effect on cultured BC cells. Furthermore, resistance to HO-5114 treatment did not differ markedly between the HR+ and TNBC lines. The latter even seemed to be more sensitive to the drug in models involving long-term treatment; however, in vitro cell culture effects translate poorly to human therapy. Accordingly, to establish the therapeutic potentiality of HO-5114, follow up experiments have to be performed in animal models for determining its in vivo toxicity and anti-neoplastic effectiveness.

## 4. Materials and Methods

### 4.1. Reagents

Hexadecyl (1-oxyl-2,2,5,5-tetramethyl-2,5-dihydro-1H-pyrrol-3-yl) diphenylphosphonium bromide (HO-5114) was synthesized and purified by us (MI and TK). All other reagents were of the highest purity commercially available.

### 4.2. Cell Cultures

MCF7 and MDA-MB-231 cell lines were purchased from American Type Culture Collection (Manassas, VA, USA). Cells were grown and maintained in a humidified incubator at 37 °C with 5% CO_2_. Estrogen and progesterone receptor-positive MCF7 cells were cultured in RPMI (Biosera, Nuaille, France) supplemented with 10% fetal bovine serum (FBS). Triple-negative MDA-MB-231 cells were cultured in DMEM Low Glucose (Biosera, Nuaille, France) augmented with 10% FBS (Thermo Fisher, Life Technologies, Milan, Italy).

### 4.3. Viability Assay

Cells were seeded at a density of an 8 × 10^3^/well in 96-well cell culture plates 24 h before the treatment. After 24 h of treatment with 1, 2.5, 5, or 10 µM of HO-5114, the medium was discarded, and the cells were washed with phosphate buffered saline (PBS; Biowest, Nuaille, France) and fixed in 100 µL of a cold 10% trichloroacetic acid (TCA) solution (Sigma-Aldrich Co., Budapest, Hungary) for 30 min at 4 °C. After TCA was discarded, the cells were washed with a 1% acetic acid solution (Sigma-Aldrich Co., Budapest, Hungary) and dried overnight at room temperature. The next day, 70 µL 0.1% sulforhodamine B (SRB) (Sigma-Aldrich Co., Budapest, Hungary) in a 1% acetic acid solution was added to the wells for 20 min at room temperature. The plates were washed 5 times with a 1% acetic acid solution and dried for at least 2 h. Added to the cell was 200 µL of a 10 mM TRIS solution (Sigma-Aldrich Co., Budapest, Hungary) and the samples were incubated at room temperature on a plate shaker for 3 h. Absorbance was measured at 560 and 600 nm simultaneously using the GloMax^®^-Multi Instrument (Promega, Madison, WI, USA). OD_600_ was subtracted as the background from the OD_560_ values.

### 4.4. Flow Cytometric Analysis of Cell Death

A flow cytometry analysis was applied to quantify the ratio of live, early apoptotic, and late apoptotic/dead cell populations. The cells were seeded into 6-well plates at a starting density of 10^5^/well 24 h before they were treated with 1, 2.5, 5, or 10 µM of HO-5114 for 24 h. The FITC-Annexin V Apoptosis Detection Kit with PI (BioLegend, San Diego, CA, USA) was used to label cells according to the manufacturer’s instructions. The samples were measured with a SONY SH800 Cell Sorter (SONY Biotechnology, San Jose, CA, USA). Debris and aggregates had been eliminated by gating, and at least 20,000 single cell events were acquired per sample. The analysis was carried out with Cell Sorter Software (SONY Biotechnology, San Jose, CA, USA). Double negative (Annexin V−/PI−) cells were considered live. Annexin V positive (Annexin V+/PI−) and double positive (Annexin V+/PI+) cells were identified as early and late apoptotic, respectively. PI positive (Annexin V−/PI+) necrotic cells were not detected. They were likely eliminated during the washing steps prior to staining.

### 4.5. Measurement of ROS Production

To measure intracellular ROS production, the cells were seeded at a starting density of 1.5 × 10^4^/well into 96-well plates and were cultured for 24 h. The cells were treated with 1, 2.5, 5, 10, or 20 µM of HO-5114 in a Krebs-Henseleit solution supplemented with 10% FBS and containing dihydrorhodamine 123 (Sigma-Aldrich Co., Budapest, Hungary). ROS generation was monitored from 0 min until 4 h using the GloMax^®^-Multi Instrument (Promega, Madison, WI, USA) at respective excitation/emission wavelengths of 490/525 nm.

### 4.6. Measurement of Mitochondrial Bioenergetics

To analyze respiratory and glycolytic energy production, OCR and ECAR were measured simultaneously by a Seahorse XFp Extracellular Flux Analyzer (Agilent Technologies, Santa Clara, CA, USA). The cells were plated at a starting density of 1.5 × 10^4^/well into Seahorse XFp Cell Culture Miniplates 24 h before treatment. The medium was replaced to the Seahorse XF Assay Media (pH 7.4) containing 10 mM glucose, 2 mM L-glutamine, and 1 mM pyruvate. After measuring the basal respiration for 18 min, HO-5114 was added to the medium at a final concentration of 1 or 2.5 µM, and the cells were further incubated for 4 h. In the final 75 min of incubation, recording of OCR and ECAR was resumed, and the following modulators were injected sequentially: oligomycin (1.5 µM final concentration), FCCP (1 µM final concentration), and rotenone and antimycin A (0.5 µM final concentration each). The OCR and ECAR data were normalized to total cellular protein, which was determined by the Micro BCA Protein Assay kit (Thermo Fisher Scientific, Waltham, MA, USA).

### 4.7. Measurement of Mitochondrial Membrane Potential

The cells were seeded to glass coverslips in 6-well plates at a starting density of 1.5 × 10^5^ cells/well and were cultured for 24 h. They were treated with 1 and 2.5 µM HO-5114 for 1 or 2.5 h for the MCF7 or MDA-MB-231 lines, respectively. After treatment, the cells were washed in PBS and incubated for 15 min at 37 °C in a modified Krebs-Henseleit solution containing 100 ng/mL of the cationic carbocyanine dye JC-1 (5,5′,6,6′-tetrachloro-1,1′,3,3′tetraethylbenzimidazolylcarbocyanine iodide). Following incubation, the cells were washed once with a modified Krebs-Henseleit solution and then visualized by a Nikon Eclipse Ti-U fluorescent microscope equipped with a Spot RT3 camera using a 20× objective lens and epifluorescent illumination. The same microscopic fields were imaged with a 490 nm bandpass excitation and >590 nm (red) or <546 nm (green) emission filters, consecutively. For quantifying red and green fluorescent intensities, their respective greyscale images were normalized to three randomly chosen spots of their backgrounds. Red and green fluorescent intensities were calculated as the percentage of their sum.

### 4.8. Colony Formation Assay

The cells were seeded at a starting density of 2 × 10^3^/well into 6-well plates and were cultured for 24 h before they were exposed to 50, 75, 100, or 250 nM of HO-5114 for seven days. Then, the cells were washed with PBS and were stained with 0.1% Coomassie Brilliant blue R 250 (Merck KGaA, Darmstadt, Germany) in 30% methanol (Sigma-Aldrich Co., Budapest, Hungary) and 10% acetic acid. The tissue culture plates were imaged using a GE Healthcare ImageScanner II (AP Hungary Co., Budapest, Hungary) set for 600 dpi. The colonies were quantified using ImageJ software.

### 4.9. Measurement of Invasive Growth

To monitor the effects of HO-5114 on the growth of MCF7 and MDA-MB-231 cells, we used the xCELLigence system that allows for the real-time, quantitative analysis of adherent cells. The measurement method is based on the use of electronic microtiter plates (E-Plate^®^), in the xCELLigence Real-Time Cell Analysis (RTCA) device (ACEA Biosciences, San Diego, CA, USA); both were used according to the manufacturer’s protocol. The instrument was placed in a humidified incubator at 37 °C and 5% CO_2_. Cells were seeded at the starting density of 1 × 10^3^/well and were cultured for 24 h. Then, the cells were exposed to 75, 100, and 250 nM HO-5114 for seven days in the E-Plate^®^, during which the impedance was measured each hour.

### 4.10. Statistical Analysis

The results are presented as mean ± standard error of the mean (SEM) of at least three independent experiments. The statistical differences between the groups were analyzed by a one-way ANOVA with the Tukey post-hoc test using OriginPro^®^ software (Originlab Corp., Northampton, MA, USA). The differences among the groups were regarded as significant at *p* < 0.05.

## Figures and Tables

**Figure 1 ijms-22-09016-f001:**
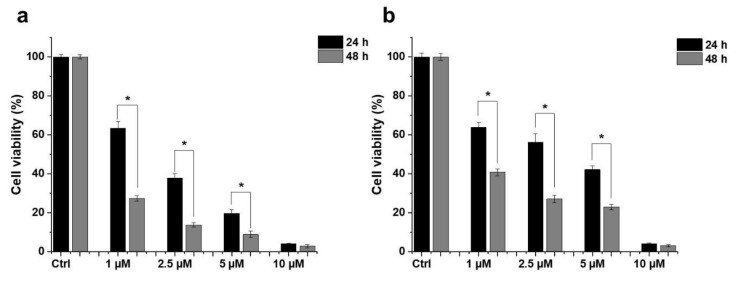
Effect of HO-5114 on the viability of human breast cancer lines. MCF7 (**a**) and MDA-MB-231 (**b**) cells were treated with 1, 2.5, 5 or 10 µM HO-5114 for 24 or 48 h, and then the viability was determined by the SRB assay. Data are shown as mean ± standard error of the mean (SEM) of at least three independent experiments running in three parallels each. * *p* < 0.05 compared to the cells treated for 24 h.

**Figure 2 ijms-22-09016-f002:**
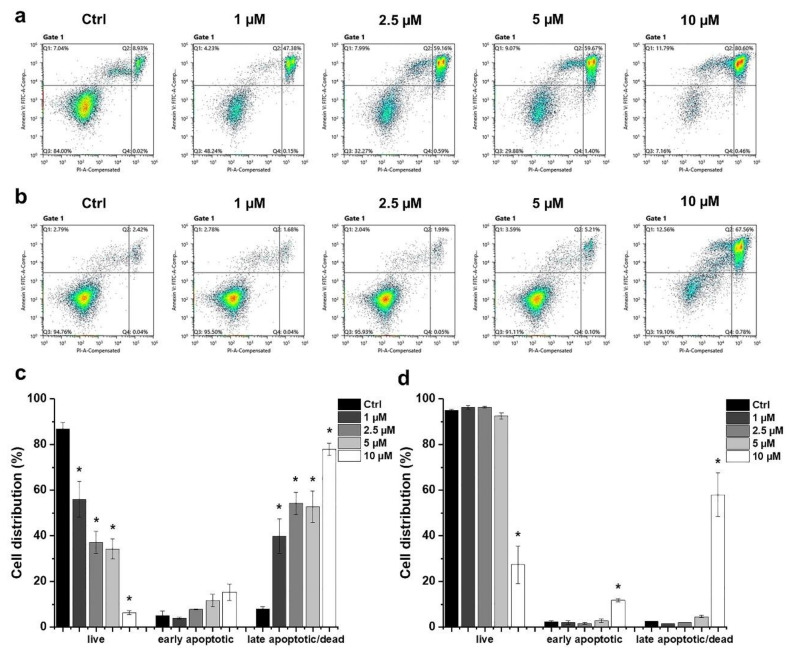
Effect of HO-5114 on the apoptosis of human breast cancer lines. MCF7 (**a**,**c**) and MDA-MB-231 (**b**,**d**) cells were treated with 1, 2.5, 5 or 10 µM HO-5114 for 24 h, and then the cells were double-stained with FITC-Annexin V and PI and were exposed to a flow cytometry analysis. Dot plots (**a**,**b**) show the distribution of early apoptotic, late apoptotic, and live cells (Q1, Q2 and Q3 quadrants, respectively). Bar charts (**c**,**d**) represent the results of at least three independent experiments. The results are shown as mean ± SEM. * *p* < 0.05 compared to the untreated cells.

**Figure 3 ijms-22-09016-f003:**
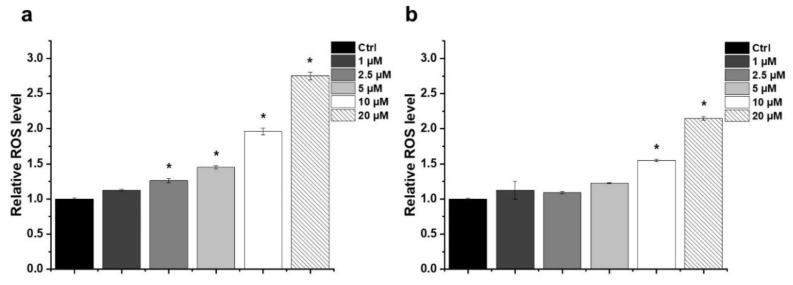
Effect of HO-5114 on cellular ROS production in human breast cancer lines. MCF7 (**a**) and MDA-MB-231 (**b**) cells were treated with 1, 2.5, 5, 10 or 20 µM HO-5114 for 4 h, and then ROS accumulation was assessed by the quantitative formation of fluorescent rhodamine 123 oxidized by the ROS from its non-fluorescent reduced precursor. The results are shown as mean ± SEM of at least three independent experiments. * *p* < 0.05 compared to the untreated cells.

**Figure 4 ijms-22-09016-f004:**
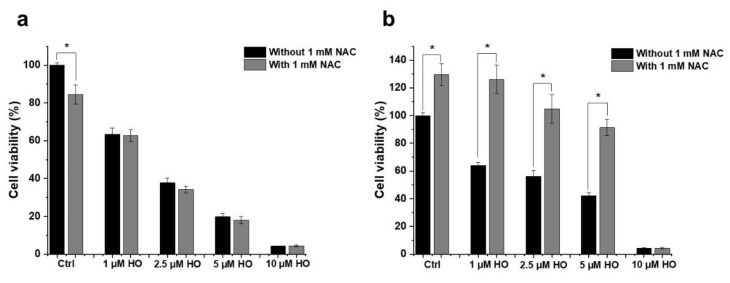
Effect of NAC on HO-5114′s cytotoxicity in human breast cancer lines. MCF7 (**a**) and MDA-MB-231 (**b**) cells were treated with 1, 2.5, 5 or 10 µM HO-5114 for 24 h in the absence or presence of 1 mM NAC, and then the viability was determined using the SRB assay. Data are shown as mean ± SEM of at least three independent experiments running in three parallels each. * *p* < 0.05 compared to the cells untreated with NAC.

**Figure 5 ijms-22-09016-f005:**
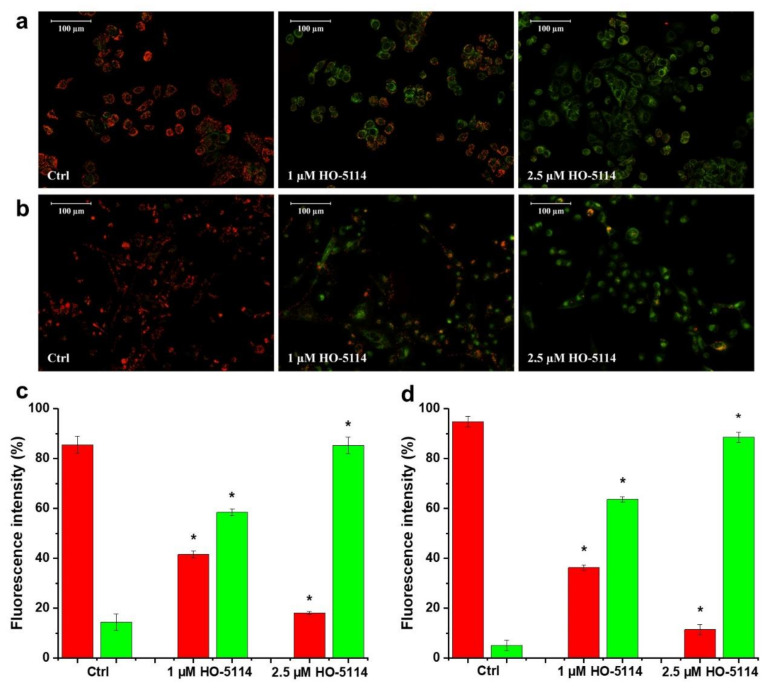
Effect of HO-5114 on the viability of human breast cancer lines. MCF7 (**a**,**c**) and MDA-MB-231 (**b**,**d**) cells were treated with 1 or 2.5 µM HO-5114 for 1 (**a**,**c**) or 2.5 (**b**,**d**) h, and then ΔΨ_m_ was assessed by fluorescence microscopy after loading the cells with the lipophilic, cationic fluorescent dye, JC-1. Red and green fluorescence indicates normal and depolarized ΔΨ_m_, respectively. Representative merged images of the same field acquired from the microscope’s red and green channels separately are presented (**a**,**b**). Quantitative assessment of ΔΨ_m_, (**c**,**d**) expressed as the % of fluorescence intensity, means ± SEM of three independent experiments. Quantitative comparisons are true within the same color only. Red and green bars denote red and green fluorescence, respectively. * *p* < 0.05 compared to the untreated cells.

**Figure 6 ijms-22-09016-f006:**
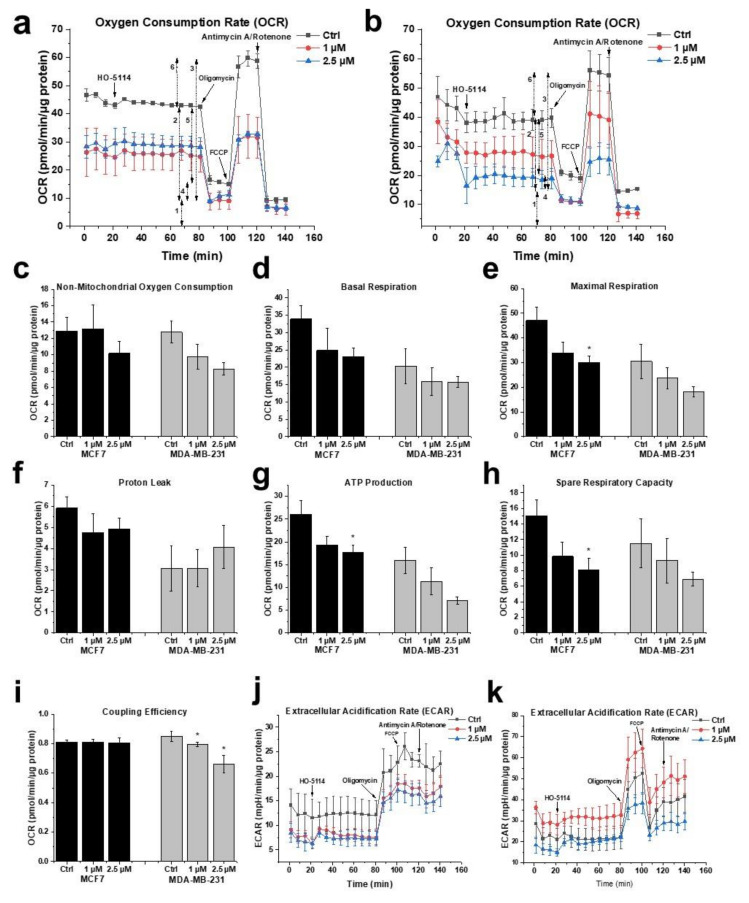
Effect of HO-5114 on the energy metabolism of human breast cancer lines. The cells were treated with 1 or 2.5 µM HO-5114 for 4 h, while OCR and ECAR were monitored during the last 75 min of treatment. The F_o_F_1_ ATP synthase inhibitor oligomycin (o), the uncoupler FCCP, and the respiratory inhibitors rotenone and antimycin A (R+AmA) were added at the bold arrows. (**a**) OCR recordings in the MCF7 line. The double-headed arrows with numbers next to them indicate: (1) basal respiration, (2) proton leak, (3) maximal respiration, (4) ATP production, (5) non-mitochondrial oxygen consumption, and (6) spare respiratory capacity. (**b**) OCR recordings in the MDA-MB-231 line. (**c**–**i**) Parameters derived from (**a**,**b**); for explanation, see the text and (**a**). (**c**) Non-mitochondrial oxygen consumption. (**d**) Basal respiration. (**e**) Maximal respiration. (**f**) Proton leak. (**g**) Mitochondrial ATP production. (**h**) Spare respiratory capacity. (**i**) Coupling efficiency. (**j**) ECAR recordings in the MCF7 line. (**k**) ECAR recordings in the MDA-MB-231 line. OCR and ECAR data were normalized to mg protein content and presented as means ± standard deviation (SD) of three independent experiments running in two parallels. * *p* < 0.05 compared to the untreated cells.

**Figure 7 ijms-22-09016-f007:**
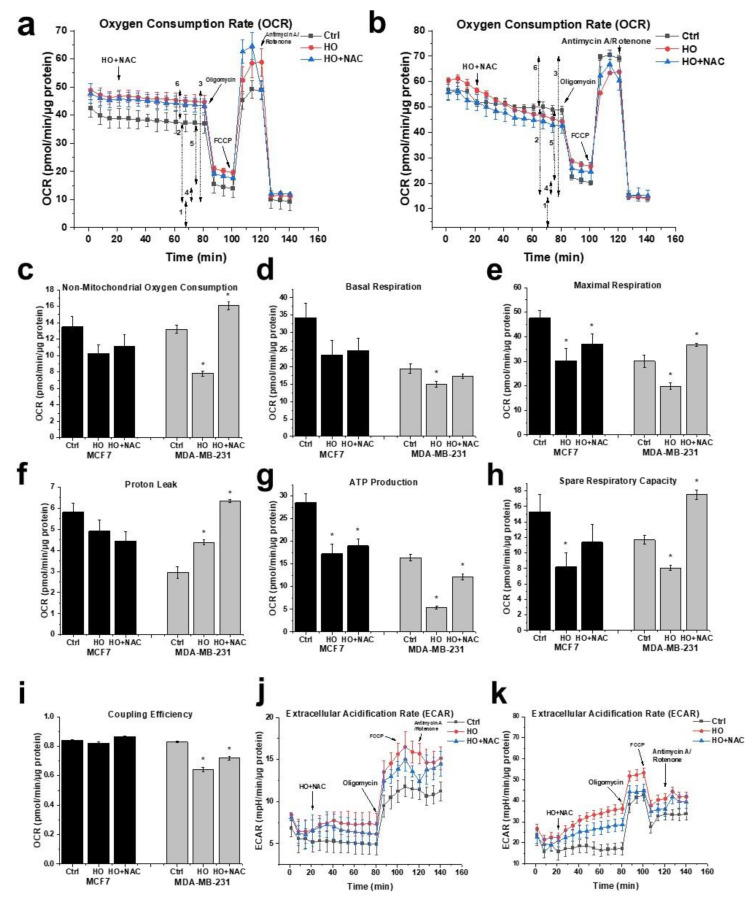
Effect of HO-5114 and NAC on the energy metabolism of human breast cancer lines. The cells were treated with 2.5 µM HO-5114 in the presence (blue line) and absence (red line) of 1 mM NAC for 4 h, while OCR and ECAR were monitored during the last 75 min of treatment. The F_o_F_1_ ATP synthase inhibitor oligomycin (o), the uncoupler FCCP, and the respiratory inhibitors rotenone and antimycin A (R + AmA) were added at the bold arrows. (**a**) OCR recordings in the MCF7 line. The double-headed arrows with numbers next to them indicate: (1) basal respiration, (2) proton leak, (3) maximal respiration, (4) ATP production, (5) non-mitochondrial oxygen consumption, and (6) spare respiratory capacity. (**b**) OCR recordings in the MDA-MB-231 line. (**c**–**i**) Parameters derived from (**a**,**b**); for explanation, see the text and (**a**). (**c**) Non-mitochondrial oxygen consumption. (**d**) Basal respiration. (**e**) Maximal respiration. (**f**) Proton leak. (**g**) Mitochondrial ATP production. (**h**) Spare respiratory capacity. (**i**) Coupling efficiency. (**j**) ECAR recordings in the MCF7 line. (**k**) ECAR recordings in the MDA-MB-231 line. OCR and ECAR data were normalized to mg protein content and presented as means ± SD of three independent experiments running in two parallels. * *p* < 0.05 compared to the untreated cells.

**Figure 8 ijms-22-09016-f008:**
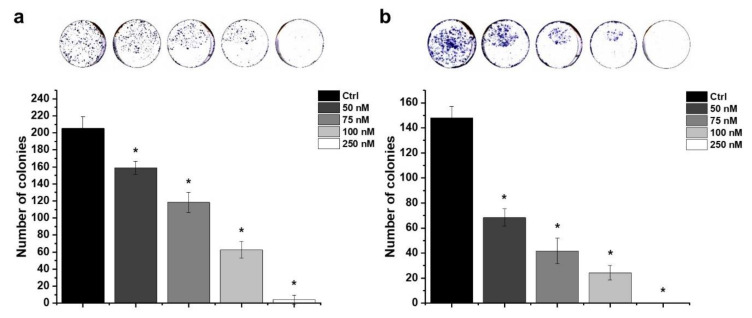
Effect of HO-5114 on the colony formation of human breast cancer lines. MCF7 (**a**) and MDA-MB-231 (**b**) cells were cultured in the presence of 0, 50, 75, 100 or 250 nM of HO-5114 for seven days and then were stained with Coomassie Blue, and the colonies were counted. The results are shown as mean ± SEM of at least three independent experiments. * *p* < 0.05 compared to the untreated cells.

**Figure 9 ijms-22-09016-f009:**
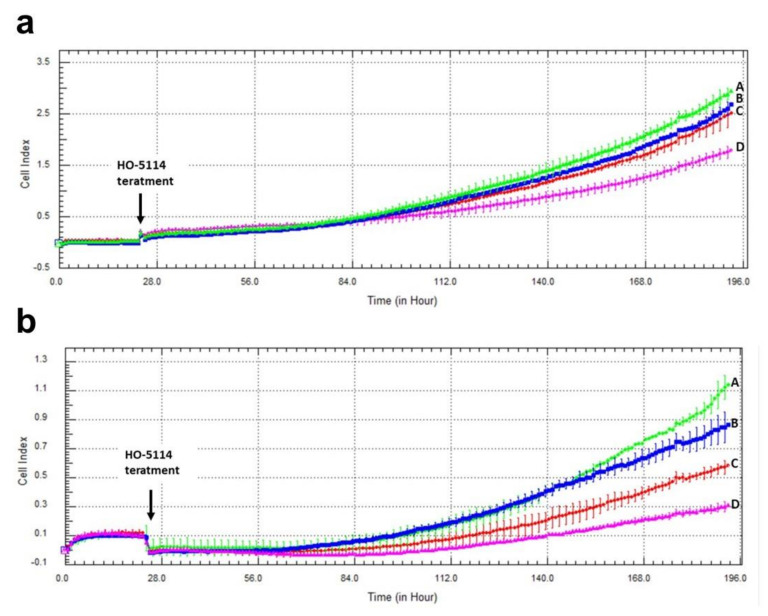
Effect of HO-5114 on the invasive growth of human breast cancer lines. MCF7 (**a**) and MDA-MB-231 (**b**) cells were cultured in the presence of 0 (line A), 75 (line B), 100 (line C), or 250 (line D) nM of HO-5114 for seven days, while the cell index was monitored in real-time. The results are shown as mean ± SEM of at least three independent experiments. * *p* < 0.05 compared to the untreated cells.

## Data Availability

All data are presented in the paper.

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
