# Peer review of "Cytostatic Effect of a Novel Mitochondria-Targeted Pyrroline Nitroxide in Human Breast Cancer Lines"

_ijms, 2021, doi:10.3390/ijms22169016_

Round 1

Reviewer 1 Report

Andreidesz et al. investigated the cellular response of breast cancer cells upon the treatment of HO-5114. The manuscript is rich in data with a few issues to be addressed. 

  1. Line 80, please provide a short reason on why SRB assay is more reliable than metabolic assay. Considering that the drug itself is a mitochondrial targeting one, it might make more sense to evaluate the metabolic activities too. 
  2. Please include details on how the quantification was conducted for Figure 4. 
  3. In the Discussion section, a conclusive paragraph is missing on the overall effect of HO-5114 on breast cancer cell line. 

Author Response

We thank the reviewer his/her thorough review of the manuscript. We believe that his/her contribution improved the manuscript considerably.

  1. Line 80, please provide a short reason on why SRB assay is more reliable than metabolic assay. Considering that the drug itself is a mitochondrial targeting one, it might make more sense to evaluate the metabolic activities too. 
    Answer: We re-phrased the sentence in line 80 that hopefully clarifies why SRB assay is more reliable for cell count than metabolic activity based assays such as MTT assay. We agree with the reviewer that it makes sense to assess metabolic activity for mitochondria targeted drugs, however, this time we were interested in cell count.
  2. Please include details on how the quantification was conducted for Figure 4.
    Answer: We added two sentences to the Materials and Methods to explain quantification.
  3. In the Discussion section, a conclusive paragraph is missing on the overall effect of HO-5114 on breast cancer cell line.
    Answer: We added a conclusion as requested by the reviewer.

Reviewer 2 Report

The manuscript by Andreidesz et al described the cytostatic effect of a novel mitochondria-targeted compound HO-5114 in two different breast cancer cell lines. Authors presented anti-neoplastic effects of HO5114 by interfering with mitochondrial respiration and membrane potential in a systematic way. However, it lacks some important controls like its effect on normal cells.
Major points:
1) Since the main challenge of drugs that target mitochondrial metabolism to the clinical trials is to establish toxicity to normal cells, authors failed to establish this by testing the effect of HO-5114 on non-transformed normal mammary epithelial cells. 
2) The time course with different concentrations of drug is missing to evaluate how effects change with time and concentration.  Since the data presented in Figs 6 & 7 indicates that lower concentrations of drug are sufficient to reduce advanced tumor growth, lower doses of drug may help in reducing toxicity to normal cells. 
3) Does anti-oxidants like NAC inhibit the HO-5114 mediated effects on mitochondrial respiration and cell death?
Minor point:
Check typos and avoid complex sentence structure (e.g. third sentence from last in the abstract)

Author Response

We thank the reviewer his/her thorough review of the manuscript. We believe that his/her contribution improved the manuscript considerably.

Major points:
1) Since the main challenge of drugs that target mitochondrial metabolism to the clinical trials is to establish toxicity to normal cells, authors failed to establish this by testing the effect of HO-5114 on non-transformed normal mammary epithelial cells. 
Answer: We agree with the reviewer that toxicity of HO-5114 on normal mammary epithelial cells could be of value for establishing a therapeutic window. However, in vitro toxicity on cell lines translate very poorly to human therapy, and for clinical trials, in vivo toxicity on normal cells and anti-neoplastic property would be much more valuable. We intend to perform these in vivo experiments, however they are out of the present study’s scope.  

2) The time course with different concentrations of drug is missing to evaluate how effects change with time and concentration.  Since the data presented in Figs 6 & 7 indicates that lower concentrations of drug are sufficient to reduce advanced tumor growth, lower doses of drug may help in reducing toxicity to normal cells. 

Answer: We fully agree with the reviewer, and added another time-point to the viability experiment. As for estimating a feasible therapeutic concentration for HO-5114, we intend to base it on in vivo toxicity and anti-neoplastic activity to be determined in a separate study.

3) Does anti-oxidants like NAC inhibit the HO-5114 mediated effects on mitochondrial respiration and cell death?
Answer: We thank the reviewer the excellent suggestion. We performed viability and energy metabolism study in the presence of NAC as requested, and added the results and their proper discussion to the manuscript.

Minor point:
Check typos and avoid complex sentence structure (e.g. third sentence from last in the abstract)
Answer: We revised the implicated sentence and made the revised manuscript proofed by Scribendi – Editing and Proofreading Services,  a commercial editing company specialized in science.

Round 2

Reviewer 2 Report

The revised manuscript reads better with improved scientific quality.